# Mortality over time among COVID-19 patients hospitalized during the first surge of the pandemic: A large cohort study

**Izabel Marcilio[1], Felippe Lazar Neto[2], Andre Lazzeri Cortez[1], Anna Miethke-Morais[1], Hillegonda Maria Dutilh Novaes[3], Heraldo Possolo de Sousa[2], Carlos Roberto Ribeiro de Carvalho[4], Anna Sara Shafferman Levin[1], Juliana Carvalho Ferreira⦿[4]\*, Nelson Gouveia[3], HCFMUSP COVID-19 Study Group[1]¶**

1 Hospital das Clinicas da Faculdade de Medicina da Universidade de Sao Paulo (HCFMUSP), São Paulo, Brazil, 2 Emergency Department, Hospital das Clínicas da Faculdade de Medicina da Universidade de Sao Paulo, São Paulo, Brazil, 3 Faculdade de Medicina, Universidade de Sao Paulo, São Paulo, Brazil, 4 Divisao de Pneumologia, Instituto Do Coracao, Hospital das Clinicas da Faculdade de Medicina da Universidade de Sao Paulo (HCFMUSP), Sao Paulo, Brazil

¶ Membership of the HCFMUSP COVID-19 Study Group is provided in the Acknowledgments.
\* juliana.ferreira@hc.fm.usp.br

**Data Availability Statement:** Our institution has an Institution-wide data management plan for COVID-19 datasets which includes making anonymized

## Abstract

### Background

Capacity strain negatively impacts patient outcome, and the effects of patient surge are a continuous threat during the COVID-19 pandemic. Evaluating changes in mortality over time enables evidence-based resource planning, thus improving patient outcome. Our aim was to describe baseline risk factors associated with mortality among COVID-19 hospitalized patients and to compare mortality rates over time.

### Methods

We conducted a retrospective cohort study in the largest referral hospital for COVID-19 patients in Sao Paulo, Brazil. We investigated risk factors associated with mortality during hospitalization. Independent variables included age group, sex, the Charlson Comorbidity Index, admission period according to the stage of the first wave of the epidemic (early, peak, and late), and intubation.

### Results

We included 2949 consecutive COVID-19 patients. 1895 of them were admitted to the ICU, and 1473 required mechanical ventilation. Median length of stay in the ICU was 10 (IQR 5–17) days. Overall mortality rate was 35%, and the adjusted odds ratios for mortality increased with age, male sex, higher Charlson Comorbidity index, need for mechanical ventilation, and being admitted to the hospital during the wave peak of the epidemic. Being admitted to the hospital during the wave peak was associated with a 33% higher risk of mortality.

data publicly available to contribute to nationwide and international registries of COVID-19 patients according to a pre-defined schedule. HCFMUSP will participate in the COVID Brazil Data-Sharing repository coordinated by The State of São Paulo Research Foundation (FAPESP), providing open access to hospital data related to COVID-19 hospitalizations. Currently, restrictions apply to the availability of these data, which were used under license for the current study and are not yet publicly available. Anonymized data are already available from the HCFMUSP COVID-19 Steering Committee (glpi.neti@hc.fm.usp.br) upon reasonable request and with permission of HCFMUSP Ethics Review Board. The release of the data to open-access repositories will be done only after the publication and dissemination of the results of initial analyses performed by our research groups and their direct collaborators.

**Funding:** This work was supported by HCCOMVIDA crowdfunding campaign.

**Competing interests:** Dr Ferreira received speaker fees from Medtronic, outside of the submitted work. This does not alter our adherence to PLOS ONE policies on sharing data and materials.

## Conclusions

In-hospital mortality was independently affected by the epidemic period. The recognition of modifiable operational variables associated with patient outcome highlights the importance of a preparedness plan and institutional protocols that include evidence-based practices and allocation of resources.

## Introduction

Two years after the first reports of COVID-19 cases in Wuhan, China, observational studies have identified individual risk factors associated with poorer outcome among hospitalized patients, such as older age, male sex, and comorbidities [1–3]. However, most published studies are from upper-middle economy and there is little knowledge about how risk factors impact hospitalization, mortality rate, and resource use in low- and middle-income countries (LMIC) [4].

Moreover, there is limited knowledge whether mortality rates have changed over time during the pandemic [5]. Recent reports from the United Kingdom and the United States suggest that mortality decreased over time [6, 7], as health professionals improved their experience with the disease and new treatments became available [8]. On the other hand, mortality rates may increase during periods of a more intense surge of cases [3–5]. Capacity strain negatively impacts patient care and outcome, and the effects of patient surge is a continuous threat during the COVID-19 pandemic as the availability of ICU beds, equipment, and specialized staff are decisive for optimal care [9–11]. Over the months of March to August 2020, the incidence of cases in Sao Paulo, Brazil, peaked and then slowly declined, and treatments such as corticosteroids or noninvasive ventilation became more frequently used [12, 13]. However, the impact of evolving clinical experience and new treatment options combined with changes in the incidence of cases on mortality in the context of an emergency preparedness plan is unknown.

Evaluating changes in mortality patterns over time is of foremost importance for resource planning and allocation and for improving patient health outcomes. As many countries experience second or third surges in COVID-19 hospitalizations after the resuming of social distancing measures, and new variants of concern are introduced, this understanding becomes paramount. Our aim was to describe clinical conditions, outcomes and risk factors associated with mortality among COVID-19 patients and to compare mortality rates over time for patients hospitalized in a large tertiary teaching hospital in São Paulo, Brazil, from March 30 to August 31, 2020.

## Methods

We conducted a retrospective cohort study based on electronic health records (EHR) of COVID-19 related hospitalizations in the largest referral hospital for COVID-19 cases in Sao Paulo, Brazil. The research protocol was approved by the Hospital das Clinicas da Faculdade de Medicina da Universidade de Sao Paulo (HCFMUSP) ethics committee (CAAE 32037020.6.0000.0068) and informed consent was waived due to the observational nature of the study.

### Study setting

Our hospital is a tertiary teaching hospital in Sao Paulo, usually dedicated to treating high-complexity cases. It comprises approximately 2,400 beds and 20,000 healthcare personnel. Its

main building, the Central Institute, operates with 900 beds, including an 84-bed intensive care unit (ICU), and a busy emergency department.

As part of the Emergency Operational Plan put in practice by the state's Health Department, the Central Institute was designated to assisting COVID-19 patients only. Its ICU capacity was increased by about 4-fold by converting regular wards to ICUs, reaching a total number of 300 ICU beds during the peak of the first wave of the epidemic. The number of available ICU beds was managed as per patient demand and staff availability during the crisis period Patient care followed institutional protocols developed specifically for COVID-19 patients, including the use of personal protective equipment, ventilatory management with low tidal volumes and protocolized sedation. Specific drugs for treating COVID-19 were not recommended but could be used at the discretion of the attending physicians. Dexamethasone was used for most patients after the publication of a clinical trial showing benefit in mid-June [8].

## Participants and data collection

We included all patients 18 years old and over consecutively admitted to the HCFMUSP from March 30 to August 31 with severe acute respiratory syndrome and a COVID-19 diagnosis confirmed either by real-time polymerase chain reaction (RT-PCR) or serology [14–16]. RT-PCR was the preferred method for diagnosis. Serology was used as a confirmatory test for probable COVID-19 cases for whom an RT-PCR test collected up to the 10th day of symptoms onset was not available. A probable COVID-19 case was defined as a suspected case (patients presenting with severe acute respiratory syndrome, defined as individuals with acute upper respiratory symptoms and presenting with: dyspnea/respiratory discomfort or persistent pressure or pain in the chest or O2 saturation below 95% in room air or cyanosis of the lips or face) and a chest Computed Tomography or chest x-ray suggestive of COVID-19 [17]. Patients who were admitted in palliative care, patients with nosocomial COVID-19, defined as patients admitted to the hospital complex for other causes, who were infected with SARS-Cov-2 during their hospitalization, and patients transferred to other facilities during their care were excluded from our sample (Fig 1).

A database of all COVID-19 cases admitted during the study period was obtained from the hospital information system. Data from each patient was then collected from the EHR and compiled onto a standardized web-based platform (REDCap–Research Electronic Data Capture) by a trained team [18]. Collected data included demographic characteristics, presenting symptoms, comorbidities, lab results, use of medications, mechanical ventilation and other resources, and clinical outcome including death, discharge, or transfer to other health facility.

## Statistical analysis

We used frequency and proportions for categorical variables, and median and interquartile range (IQR) for continuous variables. T-tests, Wilcoxon-rank sum and chi-square tests were performed according to variable distribution and type. Percentage of missing data for each variable is presented on S1 Table in S1 File. Because of the few missing values, we opted to use a complete-case analysis. No censoring was necessary once all patients admitted during the study period had been discharged when data analysis was performed.

First, we investigated risk factors associated with overall mortality during hospitalization with a logistic regression model framework: we first conducted a univariate logistic regression analysis, and then fitted a multivariate logistic regression including those independent variables with a resulting odds ratio (OR) with a p-value less than 0.2 from the univariate models. Independent variables included: age group, sex, the Charlson Comorbidity Index, admission

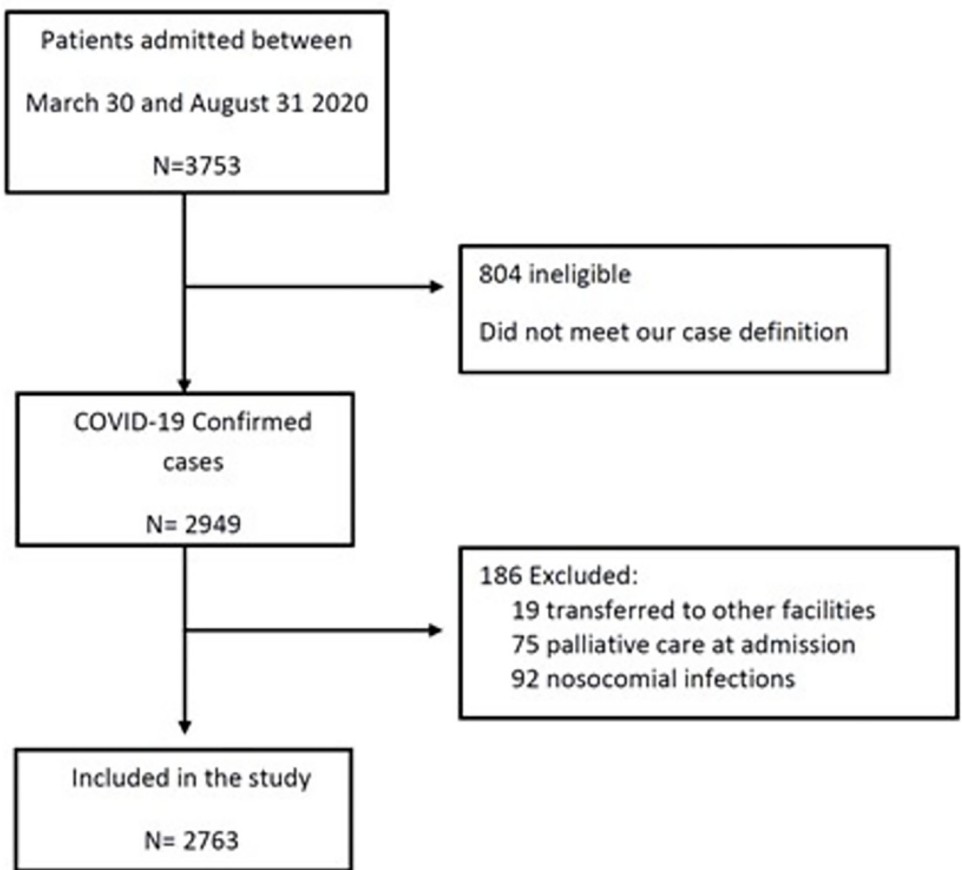

**Fig 1. Study participant flow.** Flow of potentially eligible participants in the study, and final numbers included and analyzed.

period in relation to the epidemic (early, peak, and late), time from symptoms onset, and mechanical ventilation during hospitalization. These variables were chosen to represent both patient intrinsic (age, sex, comorbidities) and disease-severity mortality risk factors (need of mechanical ventilation, days of symptoms onset) and were selected based on prior published literature.

Second, we assessed the impact of the epidemic period on mortality rates. We sub-divided the study timeframe (6-month) in three time-periods according to the stage of the first wave of the epidemic: early (March to April); peak (May to June); and late (July to August). These periods represented the different capacity strains the Hospital faced (S1 Fig in S1 File). We then modeled both the risk of overall mortality and the probability of a 30-day mortality rate during hospitalization against the period in which the patient was admitted using a logistic regression model. We adjusted for age, sex, the Charlson Comorbidity Index and need for mechanical ventilation, all known risk factors for COVID-19 mortality. Statistical analyses were done in R version 3.6.2 [19].

## Results

A total of 3753 patients were admitted between March 30 and August 31, of whom 2949 were confirmed cases according to our eligibility criteria. We excluded 19 patients transferred to other health facilities, 75 patients in palliative care at admission and 92 patients with

**Table 1. Patient characteristic according to period of admission.**

| | Total (N = 2763) | Early (N = 612) | Peak (N = 1610) | Late (N = 541) |
|---|---|---|---|---|
| **Age,** *Median (IQR)* | 61.3 (48.0, 71.1) | 58.8 (46.6, 70.5) | 61.2 (48.2, 70.7) | 63.4 (49.3, 72.3) |
| **Age Groups (Years)** | | | | |
| 18–49 | 777 (28.1%) | 189 (30.9%) | 447 (27.8%) | 141 (26.1%) |
| 50–59 | 512 (18.5%) | 133 (21.7%) | 296 (18.4%) | 83 (15.3%) |
| 60–69 | 707 (25.6%) | 130 (21.2%) | 433 (26.9%) | 144 (26.6%) |
| 70–79 | 513 (18.6%) | 103 (16.8%) | 291 (18.1%) | 119 (22.0%) |
| 80+ | 254 (9.2%) | 57 (9.3%) | 143 (8.9%) | 54 (10.0%) |
| **Male Sex** | 1557 (56.4%) | 355 (58.0%) | 901 (56.0%) | 301 (55.6%) |
| **Admission Place** | | | | |
| General Wards | 1301 (47.1%) | 333 (54.4%) | 793 (49.3%) | 175 (32.3%) |
| Intensive Care Unit | 1449 (52.4%) | 276 (45.1%) | 810 (50.3%) | 363 (67.1%) |
| Emergency | 13 (0.5%) | 3 (0.5%) | 7 (0.4%) | 3 (0.6%) |
| **Peripheral Oxygen Saturation at Admission,** *Median (IQR)* | 94.0 (91.0, 96.0) | 94.0 (90.0, 96.0) | 94.0 (91.0, 96.0) | 94.0 (92.0, 97.0) |
| **Health Care Worker** | 71 (6.0%) | 17 (14.7%) | 34 (5.0%) | 20 (5.2%) |
| **Smoking** | | | | |
| Current Smoker | 174 (6.3%) | 42 (6.9%) | 111 (6.9%) | 21 (3.9%) |
| Previous Smoker | 574 (20.9%) | 139 (22.8%) | 356 (22.2%) | 79 (14.6%) |
| **Pregnancy** | 133 (11.0%) | 18 (7.0%) | 74 (10.4%) | 41 (17.1%) |
| **CCI,** *Median (IQR)* | 2.0 (1.0, 3.0) | 2.0 (1.0, 3.0) | 2.0 (1.0, 3.0) | 1.0 (1.0, 2.0) |
| **Hospitalization Days,** *Median (IQR)* | 12.0 (7.0, 22.0) | 12.0 (7.0, 23.0) | 13.0 (7.0, 21.0) | 11.0 (7.0, 20.8) |
| **Admission to ICU** | 1895 (68.6%) | 406 (66.3%) | 1080 (67.1%) | 409 (75.6%) |
| **ICU Days,** *Median (IQR)* | 10.0 (5.0, 17.0) | 9.0 (4.2, 18.0) | 10.0 (5.0, 18.0) | 9.0 (5.0, 15.0) |
| **Maximum Oxygen Support** | | | | |
| Room Air | 267 (10.0%) | 46 (7.7%) | 177 (11.0%) | 44 (9.2%) |
| Nasal Cannula | 599 (22.3%) | 156 (26.0%) | 355 (22.1%) | 88 (18.5%) |
| Mask | 182 (6.8%) | 34 (5.7%) | 131 (8.2%) | 17 (3.6%) |
| Non-Invasive Ventilation | 107 (4.0%) | 21 (3.5%) | 80 (5.0%) | 6 (1.3%) |
| HFNC | 53 (2.0%) | 13 (2.2%) | 31 (1.9%) | 9 (1.9%) |
| Mechanical Ventilation | 1473 (54.9%) | 331 (55.1%) | 830 (51.7%) | 312 (65.5%) |
| **Days on Mechanical Ventilation,** *Median (IQR)* | 10.0 (5.0, 17.0) | 10.0 (5.0, 18.0) | 10.0 (6.0, 17.0) | NA |
| **ECMO** | 11 (0.4%) | 4 (0.7%) | 5 (0.3%) | 2 (0.4%) |
| **Dialysis** | 638 (23.1%) | 126 (20.6%) | 383 (23.8%) | 129 (23.8%) |
| **Blood Transfusion** | 561 (20.3%) | 119 (19.4%) | 340 (21.1%) | 102 (18.9%) |
| **Overall Mortality Rate** | 956 (34.6%) | 199 (32.5%) | 558 (34.7%) | 199 (36.8%) |

Charlson comorbidity index (CCI); intensive care unit (ICU); high flow nasal cannula (HFNC); mechanical ventilation (MV); extracorporeal membrane oxygenation (ECMO).

nosocomial COVID-19 infection. Therefore, 2763 patients were included in our study (Fig 1). Median age was 61 (IQR 48–71) and 56% were male. There were 71 health care workers and 133 (11%) pregnant women. Table 1 shows sociodemographic and clinical characteristics of patients.

Dyspnea (74%) and cough (68%) were the most frequent symptoms, and hypertension (58%) was the most frequent comorbidity at presentation. The median Charlson Comorbidity Index was 2 (1–3), and 217 (8%) patients were transitioned to palliative care at some point during hospitalization.

The median interval from symptoms onset to hospital admission was 7 days (IQR 5–11). 1449 of them were sent to the ICU at admission, while 1895 (69%) were admitted to the ICU at

**Table 2. Variables associated with mortality in univariate and multivariate models.**

| | | Univariable | | | Multivariable | |
|---|---|---|---|---|---|---|
| | N | OR | 95% CI | N | OR | 95% CI |
| **Period of Admission** | 2763 | | | 2331 | | |
| Early | | Reference | | | Reference | |
| Peak | | 1.10 | (0.90–1.34) | | 1.33 | (1.02–1.73) |
| Late | | 1.21 | (0.95–1.54) | | 1.00 | (0.71–1.40) |
| **Age Group (Years)** | 2763 | | | | | |
| 18–49 | | Reference | | | Reference | |
| 50–59 | | 2.25 | (1.71–2.96) | | 1.98 | (1.42–2.78) |
| 60–69 | | 4.13 | (3.24–5.30) | | 2.96 | (2.18–4.03) |
| 70–79 | | 5.58 | (4.30–7.27) | | 5.50 | (3.92–7.78) |
| ≥ 80 | | 6.82 | (4.98–9.38) | | 15.15 | (9.32–25.05) |
| **Male Sex** | 2763 | 1.51 | (1.29–1.77) | | 1.41 | (1.14–1.76) |
| **Charlson Comorbidity Index** | 2641 | 1.20 | (1.14–1.26) | | 1.26 | (1.17–1.35) |
| **Mechanical Ventilation** | 2420 | 23.27 | (17.55–31.46) | | 38.66 | (27.41–55.90) |
| **Symptoms Onset** | 2761 | 0.99 | (0.98–1.00) | | | |

Odds ratio (OR), confidence interval (CI)

some point during hospitalization. Median length of stay (LOS) in the ICU was 10 (IQR 5–17), 1473 (55%) patients required mechanical ventilation and median duration of mechanical ventilation was 10 days (IQR 5–17) (Table 1).

The overall mortality rate was 34.6%, and among the 956 patients who died during hospitalization, 88% died within 30 days. Hospital LOS was 12 (7–21) days for patients discharged alive, and 13 (8–23) days for non-survivors. S1 Table and S2 Fig in S1 File. show the characteristics of patients according to outcome.

In the univariate logistic regression analysis, older age, male sex, comorbidities (measured by the Charlson Comorbidity index), need for mechanical ventilation, and being admitted to the hospital during the peak and late periods were associated with mortality. In the multivariate logistic regression analysis, the adjusted odds ratios (OR) for mortality increased with age (Table 2). Risk of death was also increased among male patients, and among those with higher Charlson Comorbidity Index. Being admitted to the hospital during the peak period of the epidemic was associated with a 33% higher risk of mortality (OR: 1.33 (IC95% 1.02–1.73) (Table 2). We found no differences in mortality risk between early and late period of admission.

Fig 2 shows the mortality probability according to the period of hospital admission adjusted for age, sex, and Charlson Comorbidity Index. For all categories, there is a higher mortality probability for patients admitted during the peak period of the epidemic. S3 Fig in S1 File shows the increase in corticosteroids prescription over time.

## Discussion

In this cohort study, we analyzed data from 2763 COVID-19 patients admitted to a large tertiary teaching hospital between March and August 2020 and found that overall mortality was 34.6%. Older age, male sex, comorbidities and need for mechanical ventilation were risk factors associated with death, 55% of our patients required mechanical ventilation, and 69% were admitted to an ICU. In addition, we showed that admissions during the peak of the epidemic resulted in higher mortality rate after adjustment for patient individual-level data.

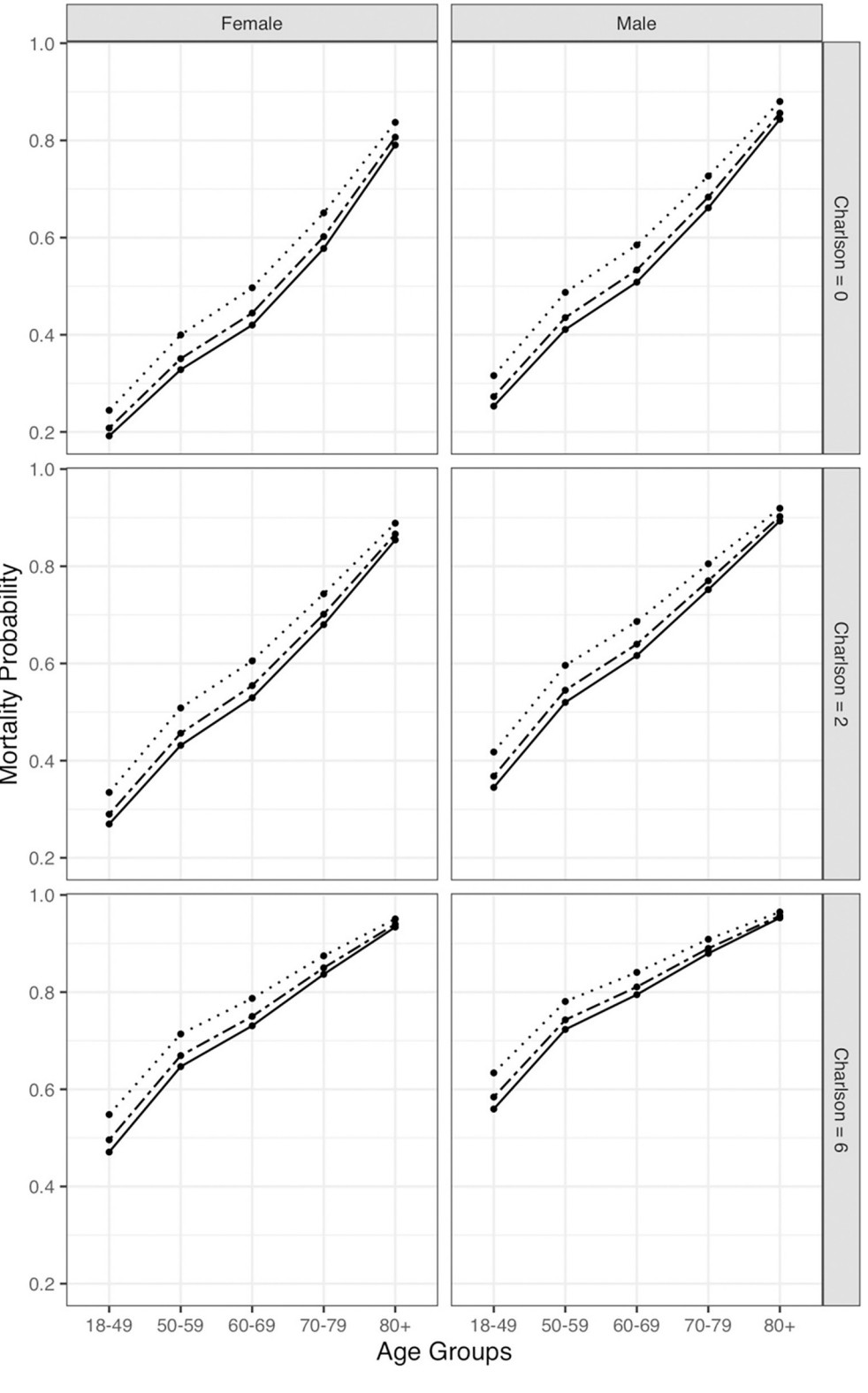

**Fig 2. Hospital mortality according to admission period.** Overall mortality probability during hospitalization according to admission period, classified as early (March and April 2020), peak (May and June 2020) and late (July and August 2020), and adjusted for need for mechanical ventilation, Charlson Comorbidity Index, sex, and age group.

This is a large cohort of hospitalized patients with COVID-19 coming from a LMIC, describing the outcomes of patients treated in a large academic hospital in the context of an emergency state preparedness plan. The plan involved designating a hospital for COVID-19 patients and transforming ward beds and operating rooms into ICU. The hospital admitted patients from all regions of the metropolitan area of Sao Paulo, which has a total population of over 21 million people. As the hospital was the main referral center for severe cases, it was constantly under strain, which may have had an important impact on mortality rate, in addition to several recognized barriers for caring for severe patients in LMIC [20].

As expected, the age structure of our cohort was younger than high-income European countries, but comparable to Israel and the United States [2, 3, 5, 21–25]. Common symptoms on admission included cough, fever, and dyspnea, and did not differ from the current literature [1, 2]. Most patients had at least one comorbidity, with a higher occurrence of hypertension and diabetes, as previously reported [1, 3, 24].

Although median LOS, median LOS in ICU, and duration of mechanical ventilation were similar to other studies, we found a higher overall mortality rate when compared to other tertiary centers in high-income countries [2, 3]. This might be due to the fact that our hospital was designated as a reference for high-severity cases only according to the state's risk-stratified Emergency Plan. We thus admitted patients referenced from secondary hospitals in the metropolitan region of Sao Paulo, which may have biased our study population towards a higher severity population. Since there was no widely used marker of disease severity at hospital admission at the time of the study, such as ISARIC 4C [26], we were unable to test the hypothesis that our patients were more severe than patients from other cohorts, or to adjust our findings to disease severity. However, we can infer increased severity from the higher rates of need for invasive mechanical ventilation, 55%, compared to less than 10% in the UK [6] and 15% in Germany [2].

When comparing the mortality rate among intensive-care units, we found a higher mortality rate than Belgium, but similar to Germany, and lower than Russia [2, 3, 27]. A recently published cohort from our hospital including only critically ill patients during a slightly different period found a 49% mortality rate [28]. Risk factors for mortality were similar to those reported previously, including age, sex, and comorbidities [1–3, 5, 28].

We provided novel data demonstrating that the mortality rate in LMIC was higher during the peak period of the epidemic, and this association held true even after adjusting for age, sex, need for mechanical ventilation and the Charlson Comorbidity Index. Recent reports have shown that increased community spread, and higher caseload are associated with worsening mortality in the United States and Israel [5, 20]. Hospital capacity strain, which often results from high community spread, has been shown to negatively impact mortality. ICU overflow has been associated with a 6% ICU mortality increase in Belgium [3], while in the United States mortality decreased from high/very high surge to low/medium surge among patients 18 to 44 years old and among those 45 to 64 years old [10].

A large nationwide study of confirmed COVID-19 hospital admissions in Brazil has found similar results [4]. This finding is possibly a result of the overwhelming burden to which health services were exposed during periods of highest community transmission of SARS-CoV-2. The COVID-19 epidemic is characterized not only by the severity of cases, but also by a rapid increase in caseload, causing a strain on health systems around the world. In order to meet the

high demand for attending critically ill COVID-19 patients, hospitals were provided with supplementary ICU beds, along with human and other resources. This rapid preparation of hospitals, however, could not prevent the overflow of patients. Thus, during peaks of caseload, hospitals could be functioning with less trained personnel, healthcare workers experiencing physical and emotional exhaustion, and temporary shortage of supplies, all of each, in turn, result in difficulties to adherence to best practices [3]. In our institution, the number of ICU beds was rapidly increased to try to meet the increased demand, but our biggest limitation was specialized teams, which had to be expanded with less experienced members during the periods of peak strain, which could have impacted clinical processes and outcomes. The additional overburden of the pandemic might have a greater impact in LMIC, where health care systems usually operate under limited resources. This could partially explain the higher in-hospital mortality found in our study [4].

We did not find a decrease in mortality rate over time as one might expect as a result of the advancing knowledge and availability of treating protocols, such as the widespread use of the prone position, use of noninvasive ventilation and the prescription of corticosteroids [6, 7, 13]. One possible explanation for this finding is that the potential benefit of a more appropriate treatment was overturned by the negative impact of patient overflow and the increased patient severity at admission.

Our study has some limitations. First, it is a single center study, in a large academic hospital with institutional treatment protocols, therefore the results may not be generalizable to other hospitals in LMIC. We collected data retrospectively, relying on information registered in EHR, which might have incompleteness or inaccuracies. Also, our population was probably biased towards high-severity cases, as admissions were managed by the State Emergency Plan, and a considerable proportion of patients were admitted under mechanical ventilation or were admitted directly to the ICU. We also acknowledge that other factors may have impacted outcomes over the time periods of the study, and for which we could not account, and therefore results may be influenced by residual confounding. We did not include any data on performance status, frailty or activities of daily living which could have impacted mortality.

This study also has important strengths. It consisted of a sizable cohort of patients admitted to the largest health care center in Brazil and one of the largest in the world dealing with COVID-19. The residential distribution of patients showed that our sample of patients was representative of the metropolitan region of São Paulo. Most importantly, previous studies of hospitalized patients with COVID-19 in Brazil were based on government records which have limited information on in-hospital trajectory, disease severity, and resource use during hospitalization [4, 12].

## Conclusion

In this large cohort study including almost 3000 patients over a 5-month period, in-hospital mortality of COVID-19 patients was independently affected by the epidemic period in the city along with well-established risk factors. The association of mortality with the overburden of health services confirms the need of public health policies to flatten the epidemic curve. Moreover, the recognition of operational variables associated with patient outcome highlights the importance of creating a preparedness plan, developing institutional protocols that include evidence-based practices and allocation of resources use to improve patient outcomes.

## Supporting information

**S1 File.**
(DOCX)

## Acknowledgments

We would like to acknowledge the outstanding work performed by healthcare workers and staff in our hospital during the COVID-19 crisis. We would also like to thank the Hospital das Clinicas COVID-19 crisis committee and the informatics department (NETI) for their support for this project. Finally, we would like to thank all members of the HCFMUSP COVID-19 Study Group: *Tarcisio E.P. Barros-Filho*[1], *Eloisa Bonfa*[1], *Edivaldo M. Utiyama*[1], *Aluisio C. Segurado*[1], *Beatriz Perondi*[1], *Amanda C. Montal*[1], *Leila Harima*[1], *Solange R.G. Fusco*[1], *Marjorie F. Silva*[1], *Marcelo C. Rocha*[1], *Izabel Cristina Rios*[1], *Fabiane Yumi Ogihara Kawano*[1], *Maria Amélia de Jesus*[1], *Esper Kallas*[1], *Maria Cristina Peres Braido Francisco*[1], *Carolina Mendes do Carmo*[1], *Clarice Tanaka*[1], *Maura Salaroli Oliveira*[1], *Thaís Guimarães*[1], *Carolina dos Santos Lázari*[1], *Marcello M.C. Magri*[1], *Julio F.M. Marchini*[1], *Alberto José da Silva Duarte*[1], *Ester C. Sabino*[1], *Silvia Figueiredo Costa*[1]

[1]Hospital das Clinicas da Faculdade de Medicina da Universidade de Sao Paulo (HCFMUSP)—Av. Dr. Enéas Carvalho de Aguiar, 255. 05403–000, São Paulo, Brazil

The lead author in the study group is Dr Aluisio Segurado (segurado@usp.br).

## Author Contributions

**Conceptualization:** Izabel Marcilio, Felippe Lazar Neto, Andre Lazzeri Cortez, Anna Miethke-Morais, Hillegonda Maria Dutilh Novaes, Heraldo Possolo de Sousa, Carlos Roberto Ribeiro de Carvalho, Anna Sara Shafferman Levin, Juliana Carvalho Ferreira, Nelson Gouveia.

**Data curation:** Izabel Marcilio, Anna Miethke-Morais, Heraldo Possolo de Sousa, Juliana Carvalho Ferreira.

**Formal analysis:** Izabel Marcilio, Felippe Lazar Neto, Andre Lazzeri Cortez, Hillegonda Maria Dutilh Novaes, Nelson Gouveia.

**Funding acquisition:** Anna Miethke-Morais, Heraldo Possolo de Sousa, Carlos Roberto Ribeiro de Carvalho, Anna Sara Shafferman Levin.

**Methodology:** Izabel Marcilio, Felippe Lazar Neto, Andre Lazzeri Cortez, Hillegonda Maria Dutilh Novaes, Heraldo Possolo de Sousa, Carlos Roberto Ribeiro de Carvalho, Juliana Carvalho Ferreira, Nelson Gouveia.

**Project administration:** Anna Miethke-Morais, Carlos Roberto Ribeiro de Carvalho, Anna Sara Shafferman Levin.

**Supervision:** Izabel Marcilio, Hillegonda Maria Dutilh Novaes, Carlos Roberto Ribeiro de Carvalho, Anna Sara Shafferman Levin, Juliana Carvalho Ferreira.

**Writing – original draft:** Izabel Marcilio, Felippe Lazar Neto, Andre Lazzeri Cortez, Juliana Carvalho Ferreira, Nelson Gouveia.

**Writing – review & editing:** Anna Miethke-Morais, Hillegonda Maria Dutilh Novaes, Heraldo Possolo de Sousa, Carlos Roberto Ribeiro de Carvalho, Anna Sara Shafferman Levin.

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
