## [Decision Letter · Decision Letter 0]

30 May 2022

PONE-D-22-12520Mortality over time among COVID-19 patients hospitalized during the first surge of the pandemic: a large cohort studyPLOS ONE

Dear Dr. Ferreira,

Thank you for submitting your manuscript to PLOS ONE. After careful consideration, we feel that it has merit but does not fully meet PLOS ONE’s publication criteria as it currently stands. Therefore, we invite you to submit a revised version of the manuscript that addresses the points raised during the review process.

We look forward to receiving your revised manuscript.

Kind regards,

Tai-Heng Chen, M.D.

Academic Editor

PLOS ONE

"I have read the journal's policy and the authors of this manuscript have the following competing interests:Dr Ferreira received speaker fees from Medtronic, outside of the submitted work. The other authors declare that they have no known competing interests."

"We thank the generous donations by companies and the general public to HC-COMVIDA crowdfunding scheme (https://viralcure.org/c/hc) managed by Fundação Faculdade de Medicina, which made this study possible."

"This work was supported by HCCOMVIDA crowdfunding campaign."

4. One of the noted authors is a group or consortium "HCFMUSP COVID-19 Study Group". In addition to naming the author group, please list the individual authors and affiliations within this group in the acknowledgments section of your manuscript. Please also indicate clearly a lead author for this group along with a contact email address.

Reviewers' comments:

Reviewer's Responses to Questions

**Comments to the Author**

1. Is the manuscript technically sound, and do the data support the conclusions?

Reviewer #1: Partly

Reviewer #2: Yes

2. Has the statistical analysis been performed appropriately and rigorously? 

Reviewer #1: Yes

Reviewer #2: Yes

3. Have the authors made all data underlying the findings in their manuscript fully available?

Reviewer #1: No

Reviewer #2: Yes

4. Is the manuscript presented in an intelligible fashion and written in standard English?

Reviewer #1: No

Reviewer #2: Yes

5. Review Comments to the Author

Reviewer #1: PONE-D-22-12520 comments

In this retrospective study from a single hospital in Sao Paulo 2949 patients with confirmed Covid-19 hospitalized in the period from March 30 and 4 months onward were studied. Data from hospital electronic records were used to assess outcome in particular hospital mortality. Overall, 35% of their patients died during hospitalization.

The study is large and gives the opportunity to study some perceived important predictive factors for mortality, but given its retrospective nature other important factors could not be studied, and the results are confirmation form other similar studies during the Covid-!9 pandemic.

Major comments.

The result presentations could be improved by dividing the patients into two cohorts: one in need of intensive care and one that could be treated in the ward. Ideally another group should also be included, patients treated in intermediate care units, but no mention are given in the manuscript to such units.

Table 1 which is an important source of information should then be divided into three cohorts: All patients, ward patients and ICU patients (both those admitted directly and during hospital stay), and the three phases could be described into this table as well. The mention of ICU patients in several places as now is a bit confusing.

I miss an important information, oxygen saturation at admission (with information of amount of oxygen require), for ICU patients this could be given as the PaO2/FiO2 ratio. This information is important to group patients according to severity at admission, also advocated by WHO guidelines.

I also would like the authors to define what definitions they use for severe acute respiratory syndrome which in addition to age (>18) and COVID-19 test positive.

You find that mortality was highest during the peak phase. I would like to see a better discussion for this finding. You managed to increase ICU beds 4-fold to 300 beds. With a median hospital LOS of 10 days this capacity over the 4 months (120 days) the hospital could theoretically have 36000 ICU days in the period but actually around 50% were used (18950 days). This means access to an ICU unit were probably not he main problem, do the authors have other explanations like quality of care and (or lack of equipment.

Other comments.

Page 5, line 99-100, please change the description of income to Upper-Middle Economies, which is according to the most recent World bank classification.

Missing important factors for mortality not included in the discussion:

Change over time:

Jung, C. et al. Differences in mortality in critically ill elderly patients during the second COVID-19 surge in Europe. Crit Care 25, 344 (2021).

Prognostic indicators found important: ADL and Frailty

Bruno, R. R. et al. The association of the Activities of Daily Living and the outcome of old intensive care patients suffering from COVID-19. Ann Intensive Care 12, 26 (2022).

Jung, C. et al. The impact of frailty on survival in elderly intensive care patients with COVID-19: the COVIP study. Critical care (London, England) 25, 149 (2021).

Reviewer #2: The title of the article is “Mortality over time among COVID-19 patients hospitalized during the first surge of the pandemic: a large cohort study”. The authors conducted a retrospective cohort study of 2949 consecutive COVID-19 patients. This study aimed to describe baseline risk factors associated with mortality among COVID-19 hospitalized patients and to compare mortality rates over time. This is an interesting paper. However, some of main important issues need to be verified to improve your work as following.

1. The authors conducted a retrospective cohort study. Please clarify censor strategies for this cohort.

2. For the multiple logistic regression, please clarify the methods for variables selection in model. For the multiple regression analysis, the prediction is their objective the model assumptions as well as model performance, a test for the interaction between variables, multicollinearity, a test for the interaction between variables, and goodness-of-fit analysis should be performed and show in the results or supplementary. On the other hand, if the association or casual inference is their aim the confounding factors according to previous knowledge should be included to the model for appropriate effect estimation.

For variable and model selection, please refer to these articles:

I. Heinze G, Wallisch C, Dunkler D. Variable selection - A review and recommendations for the practicing statistician. Biometrical J [Internet]. 2018 May 1;60(3):431–49.

II. VanderWeele TJ. Principles of confounder selection. Eur J Epidemiol [Internet]. 2019 Mar 15;34(3):211–9.

III. Steyerberg EW, Vergouwe Y. Towards better clinical prediction models: seven steps for development and an ABCD for validation. Eur Heart J [Internet]. 2014 Aug 1;35(29):1925–31. Available from: www.r-project.org

3. Include full details of how the authors handled missing data, outliers and include these in the results section. The author should elaborate about how you were dealing with that.

4. Have you tested the associations of the non-linear variables in the regression model, e.g., with splines or a polynomial?

5. Please demonstrate flowchart of participants.

6. Ultimately, the biggest drawback is the lack of practice-changing knowledge.

7. Finally, since I am not a native English user, I did not check for grammatical errors thoroughly. This should be done by an appropriate language reviewer.

6. PLOS authors have the option to publish the peer review history of their article (what does this mean?). If published, this will include your full peer review and any attached files.

Reviewer #1: No

Reviewer #2: **Yes: **Wisit Kaewput

---

## [Author Response · Author response to Decision Letter 0]

14 Aug 2022

We appreciate the reviewer's comments. We have have uploaded a word document with a point by point response

---

## [Decision Letter · Decision Letter 1]

12 Sep 2022

Mortality over time among COVID-19 patients hospitalized during the first surge of the pandemic: a large cohort study

PONE-D-22-12520R1

Dear Dr. Ferreira,

We’re pleased to inform you that your manuscript has been judged scientifically suitable for publication and will be formally accepted for publication once it meets all outstanding technical requirements.

Kind regards,

Tai-Heng Chen, M.D.

Academic Editor

PLOS ONE

Reviewers' comments:

Reviewer's Responses to Questions

**Comments to the Author**

1. If the authors have adequately addressed your comments raised in a previous round of review and you feel that this manuscript is now acceptable for publication, you may indicate that here to bypass the “Comments to the Author” section, enter your conflict of interest statement in the “Confidential to Editor” section, and submit your "Accept" recommendation.

Reviewer #2: All comments have been addressed

2. Is the manuscript technically sound, and do the data support the conclusions?

Reviewer #2: Yes

3. Has the statistical analysis been performed appropriately and rigorously? 

Reviewer #2: Yes

4. Have the authors made all data underlying the findings in their manuscript fully available?

Reviewer #2: Yes

5. Is the manuscript presented in an intelligible fashion and written in standard English?

Reviewer #2: Yes

6. Review Comments to the Author

Reviewer #2: (No Response)

7. PLOS authors have the option to publish the peer review history of their article (what does this mean?). If published, this will include your full peer review and any attached files.

Reviewer #2: **Yes: **Wisit Kaewput

---

## [Editor Report · Acceptance letter]

19 Sep 2022

PONE-D-22-12520R1 

Mortality over time among COVID-19 patients hospitalized during the first surge of the pandemic: a large cohort study 

Dear Dr. Ferreira:

I'm pleased to inform you that your manuscript has been deemed suitable for publication in PLOS ONE. Congratulations! Your manuscript is now with our production department. 

Kind regards, 

on behalf of

Dr. Tai-Heng Chen 

Academic Editor

PLOS ONE